# Increased Transmissibility of the SARS-CoV-2 Alpha Variant in a Japanese Population

**DOI:** 10.3390/ijerph18157752

**Published:** 2021-07-22

**Authors:** Hideo Tanaka, Atsushi Hirayama, Hitomi Nagai, Chika Shirai, Yuki Takahashi, Hiroto Shinomiya, Chie Taniguchi, Tsuyoshi Ogata

**Affiliations:** 1Fujiidera Public Health Center of Osaka Prefectural Government, Fujiidera 583-0024, Japan; TakahashiYu@mbox.pref.osaka.lg.jp; 2Department of Public Health and Medical Affairs, Osaka Prefectural Government, Osaka 540-8507, Japan; HirayamaA@mbox.pref.osaka.lg.jp; 3Ibaraki Public Health Center of Osaka Prefectural Government, Ibaraki 567-8585, Japan; NagaiHit@mbox.pref.osaka.lg.jp; 4Hirakata City of Public Health Center, Hirakata 573-8666, Japan; chika_shirai@city.hirakata.osaka.jp; 5Ehime Prefectural Institute of Public Health and Environmental Science, Matsuyama 790-0003, Japan; shinomiya-hiroto@pref.ehime.lg.jp; 6College of Nursing, Aichi Medical University, Nagakute 480-1195, Japan; amachi@kej.biglobe.ne.jp; 7Itako Public Health Center of Ibaraki Prefectural Government, Itako 311-2422, Japan; t.ogata@pref.ibaraki.lg.jp

**Keywords:** SARS-CoV-2, Alpha variant, N501Y, transmissibility, household contact, Japanese

## Abstract

To assess the relative transmissibility of the SARS-CoV-2 Alpha variant compared to the pre-existing SARS-CoV-2 in Japan, we performed a cross-sectional study to determine the secondary attack rate of COVID-19 in household contacts before and after the Alpha variant became dominant in Osaka. We accessed 290 household contacts whose index cases were diagnosed between 1 and 20 December 2020 (the third epidemic group), at a time when Osaka was free of the Alpha variant. We also accessed 398 household contacts whose index cases were diagnosed between 20 April and 3 May 2021 (the fourth epidemic group), by which time the Alpha variant had become dominant. We identified 124 household contacts whose index case was determined positive for the Alpha variant (Alpha group) in this fourth group. The secondary attack rates in the fourth group (34.7%) and the Alpha group (38.7%) were significantly higher than that in the third group (19.3%, *p* < 0.001). Multivariable Poisson regression analysis with a robust error variance showed a significant excess risk in the fourth group (1.90, 95% CI = 1.47–2.48) and the Alpha group (2.34, 95% CI = 1.71–3.21). This finding indicates that the SARS-CoV-2 Alpha variant has an approximately 1.9–2.3-fold higher transmissibility than the pre-existing virus in the Japanese population.

## 1. Introduction

Japan faced the fourth wave of the COVID-19 epidemic caused by infection with the novel Corona virus SARS-CoV-2 from April to May 2021 [1]. At this time, across the country, the pre-existing virus was rapidly replaced by the SARS-CoV-2 variant known as Alpha or B.1.1.7 [2] in many areas [3]. The Osaka Prefectural Government monitored the expansion of this new variant and reported that the Alpha variant positive rate among COVID-19 PCR positive samples increased from 28% in the week 1–7 March [4] to 83% in the week 26 April–2 May [3]. Previous ecological studies estimating the reproductive numbers conducted in England and the UK showed that the SARS-CoV-2 lineage B.1.1.7 (variant Alpha), which has the N501Y mutation, has a 35% to 100% higher transmissibility than the pre-existing SARS-CoV-2 [5,6,7,8]. In the Osaka Prefecture (population 8.8 million), we also experienced a marked increase in the weekly peak incidence of COVID-19, from 43.5 per 100,000 in the week 7–13 January (third epidemic wave) to 89.7 in the week 23–29 April (fourth epidemic wave) [9]. Therefore, we hypothesized that this increase was attributed to a higher transmissibility of the Alpha variant than the pre-existing SARS-CoV-2 in Japan, at a time when the transmissibility of this variant had not been well documented in an Asian population.

The present study aimed to quantify the relative transmissibility of the SARS-CoV-2 Alpha variant compared with the pre-existing SARS-CoV-2 in the setting of household contacts in the Osaka population before vaccination against COVID-19 became available to the public.

## 2. Materials and Methods

### 2.1. Subjects

When a Public Health Center (PHC) in Japan received an incidence report of COVID-19 from a physician, the PHC immediately performed contact tracing of the index case to identify persons with whom that person was in close contact during their infectious period. All the individuals who had close contact with the index case were tested by COVID-19 PCR at least once during a 14-day observation from the time of diagnosis of that case. For this study, we selected COVID-19 index cases who were diagnosed between 1 and 20 December 2020 and their household contacts (the 3rd epidemic group), and those who were diagnosed between 20 April and 3 May 2021 and their household contacts (the 4th epidemic group) from three different PHCs (total residents: 1.16 million) in the Osaka Prefecture.

Informed consent was not required from patients and contacts because active epidemiological investigation data analyses were performed in accordance with the Infectious Diseases Control Law. The Ethics Committee of Osaka University Hospital approved this study (T20114).

### 2.2. Data Items

We collected index cases’ demographic data, date of onset, occurrence of high fever (38.0 °C and over), date of diagnosis and test results for the Alpha variant. Data on the household contacts were obtained following contact tracing of the corresponding index cases. The PHC obtained PCR positive results of the household contacts even when the first test was negative, but subsequent tests received in the clinic were positive from the incidence report documented by physicians.

### 2.3. Determination of the SARS-CoV-2 Alpha Variant

The Osaka Prefectural Government started to monitor the spread of the Alpha variant using randomly selected SARS-CoV-2 positive samples at the beginning of March 2021 (testing approximately 30% of these). Identification of the N501Y was performed by real-time one-step RT-PCR assays following the guideline of the National Institute of Infectious Diseases, Tokyo. The determination of the N501Y mutation focused on the point mutation from 23,102 A (wild type) to T using two probes: 501Y-1c_VIC-MGB and N501-1c_FAM-MGB.

The Osaka Prefectural Government randomly selected 137 samples from 3406 positive N501Y mutations determined between 20 April and 3 May 2021 to be examined via whole-genome sequences of the strain at the National Institute of Infectious Diseases, Tokyo. The National Institute confirmed that all 137 samples were positive for the SARS-CoV-2 Alpha variant. Therefore, we determined the SARS-CoV-2 Alpha variant via the determination of the N501Y mutation in this setting.

### 2.4. Statistical Methods

We calculated the household secondary attack rate by analyzing the characteristics of the contacts including the diagnostic period of the index cases (3rd epidemic/4th epidemic). A Chi-square test was used for each variable to assess the influences on the secondary attack rate. A multivariable Poisson regression model with a robust error variance [10] was used to compare the rate between the 3rd and the 4th epidemic groups. To quantify the relative transmissibility of the Alpha variant with the pre-existing SARS-CoV-2, we calculated the risk ratio for the secondary attack rate in the 4th group where the index cases were Alpha positive (Alpha group) to the 3rd epidemic group with the pre-existing virus using a Poisson regression model. Possible confounding factors were the contacts’ ages (0–29, 30–59, 60 and over), their relationship to the index case (spouse or others) and the index cases’ fever (at least 38.0 °C), which were adjusted for the analysis.

## 3. Results

We extracted anonymized data on 290 household contacts of 139 index cases during the third epidemic wave and 398 contacts of 168 index cases during the fourth epidemic wave. There were 124 household contacts with the 54 index cases of people infected with the Alpha variant in the latter.

Table 1 shows the secondary attack rate among a total of 687 household contacts according to their characteristics. Individuals aged 60 and over were most susceptible to SARS-CoV-2 infection (38.8%). Spouses of the index cases showed a significantly higher secondary attack rate (42.1%) than the other contacts. The presence of a high fever (at least 38.0 °C) in the index case was significantly correlated with a secondary attack rate among household contacts (33.2% vs. 21.7%).

The household contacts of cases in the fourth epidemic group were significantly more likely to become infected than those in the third epidemic group (34.7% vs. 19.3%, *p* < 0.001). The Alpha group showed a significantly higher rate than those in the third epidemic group (38.7% vs. 19.3%, *p* < 0.001) (Table 1).

In the multivariable regression analysis, the secondary attack rate in the fourth epidemic group was significantly higher than the third epidemic group (risk ratio: 1.90, 95% confidence interval; 1.47–2.48, *p* < 0.001) (Table 2). A significant excess risk in the Alpha group was observed in comparison with the third epidemic group (risk ratio: 2.34, 95% CI; 1.71–3.21, *p* < 0.001) (Table 3).

## 4. Discussion

Our results revealed that household contacts whose index cases were diagnosed with COVID-19 in the week 20 April to 3 May 2021, when the SARS-CoV-2 Alpha variant was dominant in Osaka, had a 1.9-fold higher secondary attack rate than those whose index cases were diagnosed in the week 1 to 20 December 2020, when Osaka was free of this variant. Moreover, contacts whose index cases were infected with this variance had a 2.3-fold higher secondary attack rate than the third epidemic group. The cumulative incidence rate of COVID-19 among the population of Osaka was still only 1.02% in 10 May 2021 (90,213 cases in a population of 8.8 million) [9]. This was before the SARS-CoV-2 vaccination became available to the general population in mid-May 2021. Therefore, the risk ratios calculated in the present analysis are considered reflective of the relative degree of increased transmissibility of the SARS-CoV-2 Alpha variant in comparison with the pre-existing SARS-CoV-2 in the Japanese population. To the best of our knowledge, this is the first report assessing the transmissibility of the SARS-CoV-2 Alpha variant in a contact tracing setting.

The N501Y mutation is a key contact residue in the receptor binding domain and enhances binding to human angiotensin-converting enzyme 2 [11]. The relative transmissibility of the SARS-CoV-2 Alpha variant was assessed by comparing the reproduction number with other lineages and was reported to range from 35 to 100%, being more transmissible in England and the UK [5,6,7,8]. The risk ratio of the Alpha group (2.34) seems to be higher than the estimated relative transmissibility in England and the UK, but these estimations were based on ecological studies. A plausible explanation of these different estimations may be that Japanese individuals had a lower likelihood of transmitting the pre-existing SARS-CoV-2 than the UK population. Of note, the secondary attack rate among household contacts in December 2020 was 19.1% in the present study, which is lower than the rate among household contacts determined in London (43%) [12] and the UK (34%) [13] in 2020. The secondary attack rates in household contacts among East Asian populations reported in 2020 were also relatively lower, such as 17% in Thailand [14], 16% in Wuhan, China [15], 12% in South Korea [16], 7% in Taiwan [17] and 19% in Japan [18]. Ethnic differences resulting in a lower susceptibility to the pre-existing virus in the Japanese population might have modified the relative transmissibility of the SARS-CoV-2 Alpha variant.

In the multivariable Poisson regression analysis, we showed that spouses had an increased risk of SARS-CoV-2 infection compared with other household contacts. This is probably because spouses had closer or longer daily contact with his/her index case than the other contacts.

Our study has some limitations. The study design was cross-sectional in order to compare secondary attack rates among household contacts between a pre- and post-Alpha variant-dominant phase. Although we adjusted for some possible confounding factors in both the index cases and contacts, other factors such as seasonality could not be adjusted for and might have biased the findings. In addition, this study did not have any information on the SARS-CoV-2 vaccination because the vaccine was not officially available to the general Japanese population until mid-May 2021. This lack of information might have resulted in an underestimation of the relative transmissibility of the SARS-CoV-2 Alpha variant in this analysis. An increasing cumulative incidence of COVID-19 in Osaka residents might also have resulted in underestimating the relative transmissibility of this variant in this setting. We used the N501Y mutation as a factor to define the Alpha variant. Other variants (P.1 and B.1.351) also contain this mutation, whereas these variants were not detected in the fourth wave in Osaka

## 5. Conclusions

This study indicates that the SARS-CoV-2 Alpha variant has an approximately 1.9–2.3-fold higher transmissibility than the pre-existing virus in the Japanese population.

## Figures and Tables

**Table 1 ijerph-18-07752-t001:** Secondary attack rate (SAR) of SARS-CoV-2 for Japanese household contacts according to their characteristics.

Characteristics		No. Household Contacts	No. SARS-CoV-2 Positive	SAR %	*p*-Value
Sex	Female	396	110	27.8%	0.754
	Male	291	84	28.9%	
	Unknown	1	0	NA	
Age	0–29 years	278	66	22.3%	0.002
	30–59 years	276	78	29.7%	
	60 years+	134	50	38.8%	
Relationship to index case	Spouse	176	74	42.1%	<0.001
	Parent	179	39	21.8%	
	Children	220	55	25.0%	
	Others	112	25	22.3%	
Fever of index case	<38.0 °C	308	67	21.7%	0.003
	≥38.0 °C	337	112	33.2%	
	Unknown	43	15	34.9%	
Time from symptom onset to diagnosis of the index case	2 days or less	331	95	28.7%	0.494
	3 to 5 days	224	56	25.0%	
	6 days and over	103	34	22.0%	
	Unknown	30	9	30.0%	
Epidemic phase based on diagnosis of index case	1 to 20 December 2020	290	56	19.3%	<0.001
	20 April to 3 May 2021	398	138	34.7%	
Viral type of index case	Pre-existing virus *	290	56	19.3%	<0.001
	Alpha variant	124	48	38.7%	

* Index cases who were diagnosed with COVID-19 from 1 to 20 December 2020 were assumed not to be infected with the variant of concern.

**Table 2 ijerph-18-07752-t002:** Factors associated with the secondary attack rate of SARS-CoV-2 for Japanese household contacts focusing on the epidemic wave using multivariable Poisson regression analysis.

Factors		Risk Ratio	*p*-Value	95% Confidence Interval		
Epidemic wave based on diagnosis of index cases	1 to 20 December 2020	ref				
	20 April to 3 May 2021	1.90	<0.001	1.47	-	2.48
Age	0–29 years	ref				
	30–59 years	1.15	0.353	0.85	-	1.56
	60 years+	1.62	0.003	1.18	-	2.23
Relationship to index case	Spouse	1.56	0.001	1.21	-	2.02
	Others	ref				
Fever of index case	<38.0 °C	ref				
	≥38.0 °C	1.51	0.002	1.17	-	1.94
	Unknown	1.50	0.078	0.96	-	2.34

**Table 3 ijerph-18-07752-t003:** Factors associated with the secondary attack rate of SARS-CoV-2 for Japanese household contacts focusing on viral type using multivariable Poisson regression analysis.

Factors		Risk Ratio	*p*-Value	95% Confidence Interval		
Viral type of index case	Pre-existing virus *	ref				
	Alpha variant	2.34	<0.001	1.71	-	3.21
Age	0–29 years	ref				
	30–59 years	1.50	0.071	0.97	-	2.34
	60 years+	2.56	<0.001	1.64	-	4.00
Relationship to index case	Spouse	1.41	0.044	1.01	-	1.98
	Others	ref				
Fever of index case	<38.0 °C	ref				
	≥38.0 °C	1.48	0.021	1.06	-	2.07
	Unknown	1.11	0.824	0.44	-	2.82

* Index cases who were diagnosed with COVID-19 from 1 to 20 December 2020 were assumed not to be infected with the variant of concern.

## Data Availability

The data presented here is not publicly available due to the privacy policy of the Osaka Prefectural Government.

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
