# Peer review of "Increased Transmissibility of the SARS-CoV-2 Alpha Variant in a Japanese Population"

_ijerph, 2021, doi:10.3390/ijerph18157752_

Round 1

Reviewer 1 Report

Increased transmissibility of the SARS-CoV-2 Alpha variant in a Japanese population

The study results should be published. The topic is relevant and very required. It contributes to better understand the possible development of SARS-CoV-2 pandemic. 

The manuscript requires minor modification:

1) The third epidemic group was assessed during 20 days (in December 2020) while the fourth group during 13 days (between April and May of 2021). This unbalanced period can cause an underestimation of household contacts and index cases in the 4th group. It should be discussed this difference of observation period for both groups. An analysis of sensitivity could help better explain if an observation bias can be considered or not.

2) It should be discussed why spouse can be in increased risk of SARS-CoV-2 infection compared to others as demonstrated by both tables 2-1 and 2-2.

3) line 127: Table 2 is referred as the table 2-2.

4) Tables 2-1 and 2-2 report a relative risk. Authors use a measure of association - the risk ratio. It could be better to unify it and to use the adjusted risk ratio (aRR) - in the text and tables.

Author Response

To Reviewer 1:

1) Reviewer 1 indicated the unbalanced period can cause an underestimation of household contacts and index cases in the 4th group. However, as the number of household contacts (study subjects) are independent from magnitude of secondary attack rate among the corresponding household contacts, it cannot generate observation bias in the calculation of risk ratio of transmissibility in this study.

2) According to the comment, we inserted the following paragraph in the 4th paragraph in Discussion in line 000 to 000.

“In the multivariable Poisson regression analysis, we showed spouse had increased risk of SARS-CoV-2 infection compared to other household contacts. This is probably because spouse had closer or longer daily contact with his/her index case than the other contacts.”

3) Thank you for the indication. I changed “Table 2” to “Table 2-2” in line 127.

4) Thank you for the suggestion. I changed “Relative risk” to “Risk ratio” in Table2-1 and Table2-2.

Reviewer 2 Report

This is a well-presented and concise study on the increased infectiousness of the SARS-CoV-2 Alpha variant. The authors used metadata and PCR results to estimate the attack rate of index cases to their household contact. In doing so the authors found an increased transmission rate (attack rate) for infections associated with the Alpha variant. The methods seem sound and the paper is well written and easy to follow.

Although, I might have missed an important aspect of the study.

The authors show the clear difference in attack rate between the 3rd and 4th wave but not between alpha and not alpha infections in the 4th wave. In the latter, only 54 of 168 index cases were positive for alpha (32%). It would be worthwhile comparing the attack rate between alpha and non-alpha variants in the 4th wave only to truly assess the higher infectivity rate of alpha. The authors say that one limitation might be the difference in seasonality between December and April. By comparing secondary attack rates within the same epidemic wave one could omit this limitation. 

I suggest including a comparison between variants within the same wave. 

Also, the authors only use the N501Y mutation as a factor to define alpha. However, other variants (P.1 and B.1.351) also contain this mutation. While I don't believe these two variants would have been prominent in Japan I think the authors should mention them in the manuscript.  

Author Response

To Reviewer 2:

1) Reviewer 2 mentioned as only 54 of 168 index case (32%) were positive for alpha, it would be worthwhile comparing the attack rate between alpha and non-alpha in the 4th wave.

              As we wrote in the line 77-79, The Osaka Prefectural Government monitored the spread of the Alpha using approximately 30% of SARS-CoV-2 positive samples in the 4th wave. Alpha variant had become dominant by the late April (as written in the line 39). Therefore, we think the readers of this manuscript can notice this comparison in the 4th wave is impossible.

             2) According to the suggestion on other variants, we added the following sentence in the last sentence of Discussion.

              “We used the N501Y mutation as a factor to define Alpha variant. Other variants (P.1 and B.1.351) also contain this mutation, whereas these variants was not detected in the 4th wave in Osaka.”